# Atmospheric Wind and Pressure-Driven Changes in Tidal Characteristics over the Northwestern European Shelf

Jack Challis [1,2,*], Déborah Idier [1], Guy Wöppelmann [2] and Gaël André [3]

1  Bureau de Recherches Géologiques et Minières (BRGM), DRP/R3C, 3 Avenue Claude Guillemin, 45060 Orléans, France; d.idier@brgm.fr
2  LIENSs, La Rochelle Université, 2 Rue Olympe de Gouges, 17000 La Rochelle, France; guy.woppelmann@univ-lr.fr
3  Service Hydrographique et Océanographique de la Marine (SHOM), 13 Rue de Châtellier, 29200 Brest, France; gael.andre@shom.fr
*  Correspondence: j.challis@brgm.fr

**Abstract:** Understanding drivers of tidal change is a key challenge in predicting coastal floods in the next century. Whilst interactions between tides and atmospheric surges have been studied, the effects of wind and pressure on tides on an annual scale over the Northwestern European shelf have not been investigated. Here, a modelling approach using the shallow water MARS model is carried out to understand and quantify meteorological effects on tidal characteristics. The model setup is validated against the GESLA 3 tide gauge database. Combined and relative influences of wind and pressure are investigated using four modelling scenarios: tide only; tide, wind, and pressure; tide and wind; and tide and pressure. Influences are investigated using a single year of tidal forcing, and across multiple years of meteorological data to examine the sensitivity to temporally changing meteorological conditions. It is found that meteorology influences tidal constituent amplitudes by $+/-1$ cm, yielding changes that may locally reach 15 cm in the predicted highest tide. Analysis of the shallow water equations show three non-linear interaction terms between tide, wind, and pressure (advective effects, quadratic parameterization of bottom friction, and shallow water effect). Part of the observed changes is shown to arise from meteorologically induced mean sea-level changes.

**Keywords:** tide; tide–meteorology interaction; hydrodynamics; modelling; European shelf

## 1. Introduction

Tides are considered a predictable phenomenon; however, it has been known for some time that tidal characteristics exhibit temporal evolution [1,2]. Studies have been carried out that quantify past changes in tidal characteristics over a range of spatial scales. In Eastport, for example, on the northeastern coast of the United States, changes in M2 amplitudes of over 2.5 cm were documented throughout 1993–2019 and changes of about 1.5 cm were also documented at Portland and Boston [3]. Cartwright [2] reported changes in constituent amplitudes at Brest between 1711 and 1959 and found a decrease in M2 amplitude of about 11 cm over this period (from 331 cm decreasing to 320 cm). Pouvreau et al. [4] extended this study, finding more complex patterns in long-term M2 changes, where a decrease of 7 cm was found to occur between 1890 and 1945, followed by a 4 cm increase from 1945 to 2005. Changes in O1 were found to be less linear over the studied period, increasing by 1.3 cm between 1711 and 1936 (from 24.1 cm to 25.4 cm), with a following decrease of 0.5 cm seen from 1936 to 1959. Challis et al. [5] investigated changes in 37 constituents over 4 tide gauges (Brest, Le Conquet, Saint-Malo, and Dunkerque). Fluctuations in tidal constituent amplitudes of up to 14.5 cm were found (occurring for the L2 constituent at Saint-Malo over the period of 1960–2021). Additionally, ranges of amplitude change of up to 9.6 cm, 3.5 cm, 1.2 cm, and 1.6 cm were found to occur for M2, N2, K1, and O1 among all

four studied sites. Regarding the quarter-diurnal constituents, in Dunkerque, changes up to 1.7 cm and 1.8 cm were found for M4 and MN4 constituents, respectively.

Observed temporal changes in tidal characteristics can be attributed to a wide range of non-astronomical drivers, affecting tides over a large range of temporal and spatial scales (see [6] and references therein). Examples of these drivers include changes in mean sea level (MSL) [7–9], wave power [10], stratification [11], riverine input [12], and basin shape changes (through dredging for example). In addition, tide and atmospheric storm surges (induced by atmospheric wind and pressure) are also known to exhibit interactions [13–22].

These interactions have been investigated in literature through numerical modelling experiments [13–20] and tide gauge data analysis [15,21,22] in many locations over the world, for instance, in Canada [16], the Patagonian Shelf [18], the Bay of Bengal [17], the Taiwan Strait [19], Tieshin Bay (China) [20], the English Channel [13], and the Irish Sea [14]. Magnitude orders are found to reach tens of cm up to 1 m depending on the location. However, the goal of these studies was primarily to quantify a combined tide–surge interaction term or to identify the effects of tides on surge magnitude and timing, rather than to quantify the effect of surge (or wind and pressure) on tide characteristics. Some of these studies (e.g., Rossiter [21]) qualitatively discuss the effect of surge on tide, highlighting that surge changes the phase of the tide. However, they do not quantify the surge effect on tides. Thus, in an effort towards a better understanding of tide changes, it appears worthwhile to investigate and quantify the sole effect of wind and pressure on tidal characteristics. Such investigation may be useful for extreme value analysis and coastal flood risk assessment. Indeed, in practice, extreme coastal water levels are estimated assuming that tidal characteristics do not change with time. Therefore, a quantification of wind and pressure-driven tidal characteristic changes may be useful to better identify the necessity (or lack thereof) of taking into account the wind and pressure-driven non-stationarity of tidal characteristics in extreme value analysis methods. The present study focuses on wind and pressure-driven changes in yearly tidal characteristics on the Northwestern European shelf.

We rely on numerical simulations over 1 year (where 2009 is chosen as a reference year, and we build a model setup, which has been previously validated against this year), accounting for hourly wind and pressure fields corresponding to different meteorological years (from 1980 to 2021). Using these simulations, we are then able to focus on three key aspects: quantification of meteorological effects on tidal constituent amplitudes, investigation of the resulting changes in the highest predicted tide, and understanding the underlying mechanisms driving these alterations. The first aspect is split into two parts. The first part involves a study into the effects of wind and pressure-driven changes on tidal characteristics over a single study year. The second part is an investigation into the sensitivity of tidal characteristics to changing meteorology using a single year of tidal forcing. The single year study of 2009 allows us to gain insight into physically realistic magnitudes of wind and pressure changes occurring over a given year, whilst sensitivity of tidal changes to changing meteorological forcing allows us to examine upper and lower limits of wind and pressure-driven effects over a long period. Investigating the resultant effects on highest tide presents benefits in the fields of coastal inundation prediction, as well as coastal erosion and statistical extremes calculation. Understanding magnitudes of changes in individual tidal constituents, and the mechanisms driving changes are both important in the field of tidal dynamics and can contribute to increased accuracy in future tidal predictions.

The paper is organised as follows. First, the model setup and simulations are described in Section 2. Then, the effects of wind and pressure change on tidal characteristics and predicted highest tide are investigated showing changes of millimetres and centimetres, respectively (Section 3). Section 3 finishes with an investigation into a potential physical mechanism responsible for wind and pressure-driven tidal characteristic changes. Discussions of additional mechanisms and comparisons with similar literature are presented in Section 4. Finally, conclusions are drawn in Section 5.

## 2. Materials and Methods

### 2.1. Model Setup and Validation

For this study, the Model for Application at Regional Scales (MARS) was selected, which is based on the shallow water equations [23]. The setup for this model builds on a configuration previously used in [24] (and later in [7]). Computations are carried out over a large area of the Northwestern European continental shelf at a 2 km resolution (inset Figure 1), with analysed outputs on a 6 km resolution over a zoomed area (Figure 1). A Strickler Coefficient of 35 $m^{1/3}s^{-1}$ is used to calculate bed shear stress. Furthermore, 14 major tidal constituents (Mf, Mm, Msqm, Mtm, O1, P1, Q1, K1, M2, K2, 2N2, N2, S2, and M4) derived from the FES2004 global tidal model [25] are used to force the model at the open boundaries. The model bathymetry used in this study is the same as that of Muller et al. [24], which was created using a combination of multiple datasets including General Bathymetric Chart of the Oceans (GEBCO; https://www.gebco.net/ (accessed on 1 December 2008)) data for deeper water areas and Lidar and echo-sounding data for shallower areas.

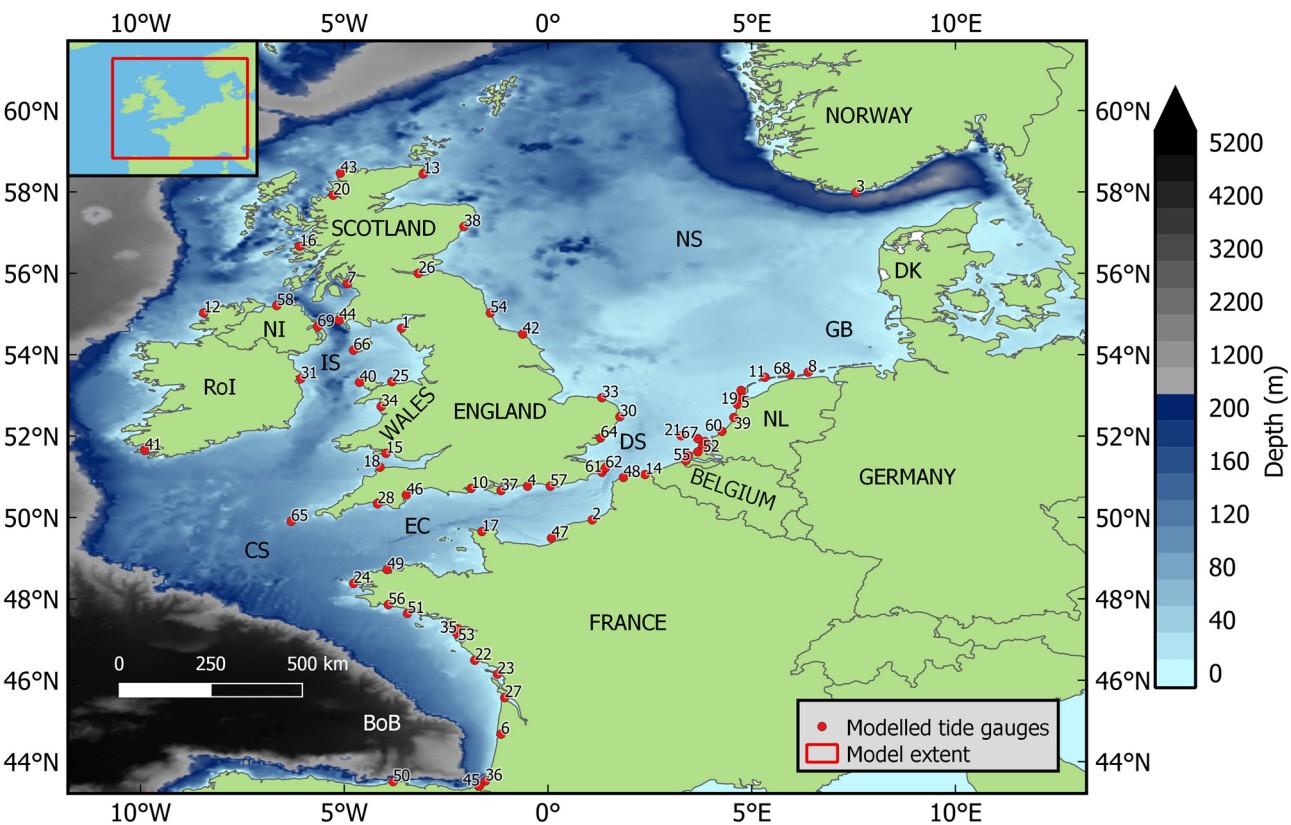

**Figure 1.** Computational domain of hydrodynamic modelling and the location of tide gauges used for validation (note that the shelf break is indicated by a change in colour scale). The inset map shows the full computational domain, whilst the main map shows the studied area. Abbreviations are as follows: BoB—Bay of Biscay, EC—English Channel, DS—Dover Strait, NS—North Sea, GB—German Bight, DK—Denmark, NL—The Netherlands, NI—Northern Ireland, RoI—Republic of Ireland, CS—Celtic Sea, and IS—Irish Sea (see Table A1 in Appendix A for names corresponding to station numbers).

For the meteorological forcing, a wind ($U$ and $V$ components for 10 m above ground; henceforth $U_{10}$ and $V_{10}$) and atmospheric pressure (atmospheric pressure at mean sea level; $P_{MSL}$) dataset was built by combining two datasets: Climate Systems Forecast Reanalysis (CFSR) [26], which runs from 1979 to 2011, and the second iteration of the CFSR dataset, CFSv2 [27], which runs from 2011 to present (where we use data up to the end of 2021). These hourly meteorological variables have original spatial resolutions of $0.312° × 0.312°$

and $0.205° \times 0.204°$ for CFSR and CFSv2, respectively. As this dataset is created by merging the CFSR and CFSv2 datasets, it will henceforth be referred to as CFSR.

Before using the model for numerical experiments on the effect of wind and pressure on tides, we may ensure that it reproduces water level storm surge and tidal dynamics in the study area to a reasonable degree. First, we should note that the model setup has been previously validated in terms of tidal dynamics at the Northwestern European continental shelf scale, considering 16 tide gauges, using a tide only simulation over the entire 2009 year, focussing on the 2009 high tide and M2, N2, K2, S2, M4 tidal constituents [7]. Storm surges have also been validated for this model setup along the French coast (4 tide gauges) on event (storms) and pluriannual (10 years) time scales, based on simulations accounting for tide and meteorological forcing [24] (but not using the same meteorological forcing data). To complete this validation in terms of total water level (i.e., accounting for tidal components and tidal residual), simulations are carried out for the year 2009, accounting for tide and meteorological forcing.

Validation is thus carried out for the total water level forced with tide, wind, and atmospheric pressure ($\zeta_{TWP}$, Figure 2a), the tide only component of the water level ($\zeta_T$, obtained through tidal prediction using constituent characteristics determined by tidal harmonic analysis (THA), Figure 2b), the tidal residual ($\zeta_{TWP} - \zeta_T$, Figure 2c), and the constituent amplitudes of six major constituents ($A_i$, Figure 3). The UTide THA software package (Python distribution, version 0.3.0) is used for validation [28], both to extract tidal constituent characteristics and to predict tidal water levels based on the estimated tidal constituent characteristics. This software is designed to analyse water level time series, even when there are missing data points (which is common in water level observations). In order to maximise the number of validation stations admitted for analysis in this study whilst also ensuring the separation of the studied constituents (K1, O1, M2, N2, M4, and MN4), tide gauges are selected that contain at least 28 days of continuous data during the 2009 period.

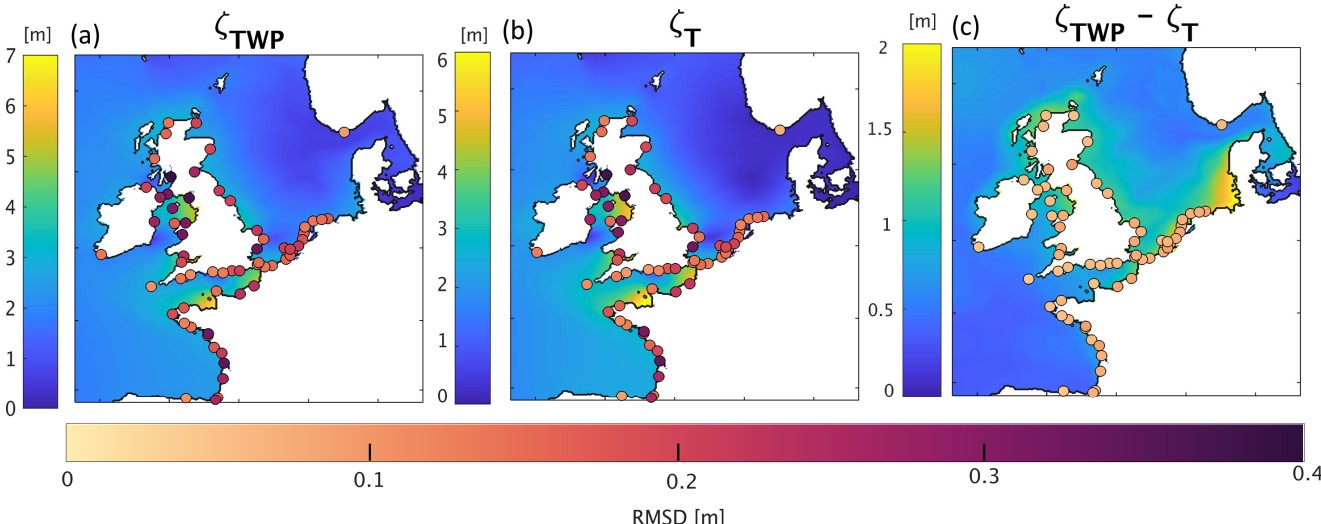

**Figure 2.** Validation results for the model configuration in terms of (**a**) water levels, (**b**) tidal component, and (**c**) tidal residual. The colour bar below indicates the RMSD (Equation (1)) for the validation stations (circles), and the blue/yellow colour scale shows the maximum values reached over the domain for the whole year. Note the different colour bar scale ranges between (**a**–**c**).

Validation results are quantified using the root mean square difference (RMSD, Equation (1)) between observational and modelled results. Note the deliberate use of the term RMSD (and not RMSE, where 'E' stands for error). Differences can arise from errors in the observations or the modelling, but also from the many coastal processes

that tide gauges can record (see Woodworth et al. [29] for a review), which this type of modelling does not intend to reproduce.

$$\text{RMSD} = \sqrt{\frac{1}{n}\sum_{i=1}^{n}|\zeta_{\text{obs}} - \zeta_{\text{mod}}|^2} \tag{1}$$

In Equation (1), *n* represents the number of temporal points in the data, where $\zeta_{obs}$ is observational data and $\zeta_{\text{mod}}$ is modelled data.

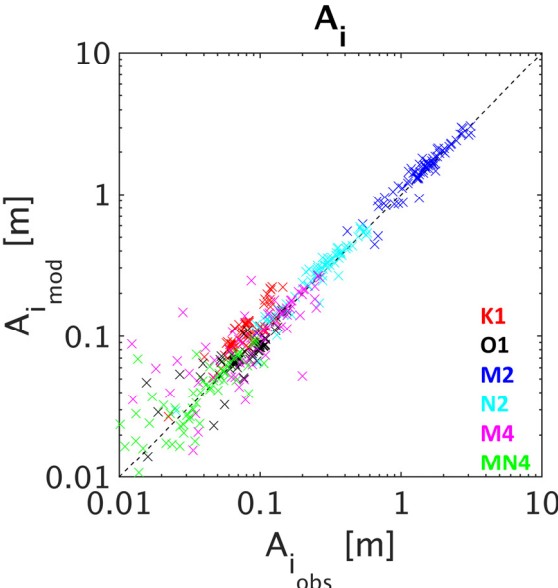

**Figure 3.** Comparison of tidal constituent amplitudes obtained from THA of observed tide gauge data (*x*-axis) and of modelled water levels (*y*-axis). R-squared (R2) values for each constituent are as follows: K1—0.73, O1—0.77, M2—0.94, N2—0.93, M4—0.66, MN4—0.71.

The modelled water levels ($\zeta_{\text{TWP}}$) exhibit a median RMSD of 18 cm (ranging from 9 to 39 cm) with respect to the observed water levels (tide gauges), with maximum values occurring at Millport (West Scotland) and minimum values occurring in Tregde (Southern Norway). The predicted tidal water levels (Figure 2b) exhibit a median RMSD of 16.5 cm (ranging from 6.9 cm to 37.5 cm, where the minimum and maximum are again found in Millport and Tregde, respectively). These values are comparable to those obtained by Pineau-Guillou [30] (where the same computational grid was used, and validation was carried out over the 2008–2009 period) with RMSD ranging from 10 to 58 cm for the French tide gauges (that are not exactly the same as in our study), and with a median RMSD value of 22 cm. Considering the tide gauges in common, we may note that, both models exhibit large RMSD values along the south–west coast of France (Arcachon-Eyrac (station 6, Figure 1) and Port Bloc (station 27, Figure 1) stations located in the Arcachon Lagoon and the Gironde estuary, respectively). Going into more detail, Figure 3 shows a fair agreement of the amplitudes of modelled tidal constituents K1, O1, M2, N2, M4, and MN4 with the observational ones, with mean differences of 3.0 cm, 0.4 cm, 6.1 cm, 1.6 cm, 0.4 cm, and 0.05 cm, respectively (where correlation coefficient values for each constituent are noted in the Figure 3 caption). For M2, N2, and M4, similar differences are found in [7], where computations were carried out with the same MARS model, but with no meteorological forcing. The study of Fernández-Montblanc et al. [31] also provides useful information for comparison for K1, O1, and M2, with amplitude differences falling in the ranges [−4 to 4], [−2.5 to 2.5], [−20 to 25] cm, respectively, on the spatial domain in common.

Regarding the modelled tidal residuals (Figure 2c), they exhibit a median RMSD of 6.7 cm (ranging from 3.8 cm to 9.1 cm). The largest RMSDs occur along the Dutch coast. Fernández-Montblanc et al. [31] also found that the RMSDs along the Dutch coast are

overall larger than in the rest of our computational domain. RMSD values in [31] were found to range from about 5 to 20 cm. Differences between these results and ours may be partly explained by a longer validation period (6 years, whilst the present work focuses on one year). Results can also be compared to those of Muller et al. [24], where RMSD values at Dunkerque (station 14 in Figure 1), La Rochelle (station 23 in Figure 1), and Le Conquet (station 24, Figure 1) over a 7 years period of validation are close to 10 cm. To avoid the time duration issue, we compared our results to the ones of [24] on the storm event occurring between the 9th and 12th of February 2009 (Storm Quentin), using the same statistics of event scale error used in [24] (maximal error and peak error). At Le Conquet, Dunkerque, and La Rochelle peak errors of 6 cm, 47 cm, and 7 cm, and maximal errors of 18 cm, 47 cm, and 61 cm were found in [24]. Similarly, we find peak errors of 11 cm, 23 cm, and 18 cm and maximal errors of 27 cm, 32 cm, and 53 cm for Le Conquet, Dunkerque, and La Rochelle, respectively.

Overall, the model set up for this study exhibits a fair agreement with the observations and existing studies, keeping in mind that it is used mainly to investigate the sensitivity to meteorological forcing conditions, rather than to provide absolute values.

*2.2. Simulations and Wind/Pressure Influence Calculations*

To investigate the effect of wind and pressure on tidal characteristics, we use the model presented above to carry out controlled numerical simulations of a reference scenario, and numerous scenarios involving differing years of meteorological forcing.

Four initial model runs were carried out for the year 2009, forced with the CFSR wind and pressure dataset, using four different sets of meteorological conditions. From here, the term scenario will refer to a single model run carried out with a specified meteorological condition (see the defined scenarios below).

- Tide only (T) scenario
- Tide and combined wind and pressure (TWP) scenario
- Tide and wind (TW) scenario
- Tide and pressure (TP) scenario

THA is carried out on the model output of each of the four aforementioned scenarios using the Tidal ToolBox (TTB) software package (2014 version) [32] to estimate tidal amplitudes for each of the two principal diurnal (K1, O1), semi-diurnal (M2, N2), and quarter-diurnal (M4, MN4) tidal constituents. This software is specifically designed to process numerical model outputs (i.e., a spatiotemporal regular grid of water levels and currents).

2.2.1. Effect of 2009 Meteorological Forcing on 2009 Tidal Constituent Amplitudes

To investigate the impact of the meteorological forcing on tidal constituent amplitudes, the following calculation (Equation (2)) is carried out taking into account both tide and meteorological forcing for the year 2009 (Y0):

$$\Delta A_{i_{TWP-T}} = A_{i_{T_{Y0}WP_{Y0}}} - A_{i_{T\,Y0}} \tag{2}$$

where $\Delta A_{i_{TWP-T}}$ denotes changes in the amplitude ($\Delta A_i$) of a tidal constituent (i) due to the combined influence of wind and pressure, in comparison to the amplitude obtained with the tide-only simulation ($A_{i_{T\,Y0}}$, where Y0 denotes the year 2009). The same equation is used to investigate relative wind and pressure-driven changes where the WP subscript is replaced with that of the corresponding scenario.

2.2.2. Sensitivity of 2009 Tidal Amplitudes to Meteorological Conditions (1980–2021)

In addition to calculating the realistic tidal response to 2009 meteorological forcing, sensitivity testing is carried out for multiple years of meteorological data. That is, the TWP scenario is realised 42 times using 2009 tidal forcing with wind and pressure forcing

data for each year between 1980 and 2021 (from here, any year (j) stated in tandem with a specified scenario refers to the year of meteorological forcing used).

These results are used to discuss the influence of temporally changing meteorology, and as such, use the 2009 TWP scenario results as a reference. The mean influence of this meteorological forcing change over these scenarios is then calculated (Equation (3)) using the following formula:

$$\overline{\Delta A_i} = \frac{1}{n}\sum\nolimits_{j=Y1}^{Y2}\left(A_{i_{T_{Y0}WP_j}} - A_{i_{T_{Y0}}WP_{Y0}}\right) \tag{3}$$

where n denotes the number of simulations carried out (in this case, 42), and j denotes the meteorological forcing year (where Y1 = 1980, and Y2 = 2021).

The standard deviation of this influence is also calculated (Equation (4)) to investigate the variability, using the following formula:

$$\sigma(\Delta A_i) = \sqrt{\frac{1}{n-1}\sum\nolimits_{j=Y1}^{Y2}\left(A_{i_{T_{Y0}WP_j}} - \overline{\Delta A_i}\right)} \tag{4}$$

Finally, the range of values of changes over all the meteorological forcing years is calculated ($R(\Delta A_i)$, Equation (5)), in order to view the full span of maximum and minimum influences.

$$R(\Delta A_i) = \Delta A_{i_{max}} - \Delta A_{i_{min}} \tag{5}$$

To distinguish between the notation used for work considering only meteorological forcing of 2009 and that considering 1980 to 2021, $\Delta A_i$ without further subscript (such as $TWP - T$ etc.) will henceforth refer to the latter (and by extension, subscripting with reference to meteorological scenarios will refer exclusively to results produced using only meteorological forcing for the year 2009).

### 2.2.3. Effect on Highest Tide

In addition to investigations on constituent amplitude, we also investigate the effect of the meteorological conditions on the highest predicted tide over the year 2009. For this purpose, TTB is used to predict the tide using the characteristics of 94 selected tidal constituents (Table A2) estimated by TTB for T and TWP. Predictions are carried out at a 5 min interval in order to correctly capture the highest tide, avoiding the issue of under sampling. As some tidal constituents are radiational or partly radiational (i.e., generated or influenced by a cyclic geophysical phenomenon rather than gravitational), it is implied that specific attention should be devoted to avoiding double-counting the effect of these radiational constituents (see Williams et al. [33]). In the present MARS model setup, the tide forcing conditions (FES2004) includes a single radiational constituent (S2). Thus, to avoid double-counting the meteorological effect in the tide prediction based on the tidal constituents coming from the TWP simulations, the S2 constituent is replaced with the S2 constituent obtained from T simulations.

Using the subscripting defined for $\Delta A_i$, we can define change in highest tide ($\Delta\zeta_{max}$, Equation (6)) as follows:

$$\Delta\zeta_{max} = \zeta_{max_{T_{Y0}WP_j}} - \zeta_{max_{T_{Y0}}} \tag{6}$$

The mean effect of meteorology on highest (predicted) tides of the year 2009 is therefore calculated using Equation (7):

$$\overline{\Delta\zeta_{max}} = \frac{1}{n}\sum\nolimits_{j=Y1}^{Y2}\left(\zeta_{max_{T_{Y0}WP_j}} - \zeta_{max_{T_{Y0}}}\right) \tag{7}$$

The standard deviation of the effect on highest tide is calculated using Equation (8):

$$\sigma(\Delta\zeta_{max}) = \sqrt{\frac{1}{n-1}\sum\nolimits_{j=Y1}^{Y2}\left(\zeta_{max_{T_{Y0}WP_j}} - \overline{\Delta\zeta_{max}}\right)} \tag{8}$$

We also investigate the largest changes in highest tide (may be positive or negative) using the following Equation (9):

$$\Theta(\Delta\zeta_{max}) = \begin{cases} \max(\Delta\zeta_{max}), & |\max\Delta\zeta_{max}| \geq |\min\Delta\zeta_{max}| \\ \min(\Delta\zeta_{max}), & |\max\Delta\zeta_{max}| < |\min\Delta\zeta_{max}| \end{cases} \tag{9}$$

This formula dictates that for areas in which the absolute of the maximum highest tide change is larger than the absolute of the minimum tide, we investigate the maximum (and conversely where the absolute of the minimum is larger, we investigate the minimum).

### 2.2.4. Influence of Meteorologically Induced MSL Changes

Finally, we analyse the influence of meteorologically induced annual MSL changes as a mechanism driving wind and pressure-induced tide changes. To investigate the part of combined wind and pressure-driven tidal characteristic changes resulting from meteorologically induced MSL, we estimate the sole effect of this meteorologically induced MSL on the tide, by using the inverse barometer equation to estimate an equivalent atmospheric pressure and run the hydrodynamic model with this equivalent atmospheric pressure. More precisely, MSL changes retrieved from a 2009 TWP scenario using THA were used in the Inverse Barometer (IB) equation (Equation (10)) (adapted from Pugh and Woodworth [34]).

$$\Delta\zeta = \frac{P_{ref} - P_n}{\rho g} \tag{10}$$

where $P_{ref}$ is the reference atmospheric pressure value used in our model setup (i.e., 1015 hPa), and $P_n$ is the equivalent atmospheric pressure. $\rho$ and $g$ are water density and gravity acceleration, respectively (where a reference seawater density value of 1027.34 $Kgm^{-3}$ is used). Equation (10) is rearranged and the term $\Delta\zeta$ replaced by $\Delta MSL$ to obtain Equation (11), allowing us to calculate the simulated equivalent pressure field, and thus investigate the effect of the MSL changes (induced by W and P) on the tide.

$$P_n = -(\Delta MSL \rho g - P_{ref}) \tag{11}$$

This pressure field is computed using the yearly MSL values obtained from the TWP computation for the year 2009, keeping in mind that MSL = 0 for the tide-only computation. This pressure field (which is invariant in time and non-uniform in space) was then used as meteorological forcing in a new scenario, including tidal forcing (TP(MSL)), where THA was then applied. To calculate the part of change exhibited in the WP caused by $MSL_{WP}$, a ratio ($\beta$) between tide changes induced by the equivalent pressure field and the ones directly induced by the effect of wind and pressure is calculated (Equation (12)).

$$\beta = \frac{\Delta A_{i_{TP(MSL)}}}{\Delta A_{i_{TWP}}} \tag{12}$$

where $\Delta$ represents tidal constituent change from the 2009 reference tide-only scenario. The closer the $\beta$ value is to 1, the greater the contribution of the wind and pressure-induced MSL changes on tidal constituent changes ($\Delta A_{i_{TWP}}$).

## 3. Results

### 3.1. Effect of 2009 Meteorological Forcing on 2009 Tidal Constituent Amplitudes

Figure 4a displays the amplitudes of each of the six studied constituents, with Figure 4b–d showing the influence of the 2009 WP, and the separate influences of W and P, respectively.

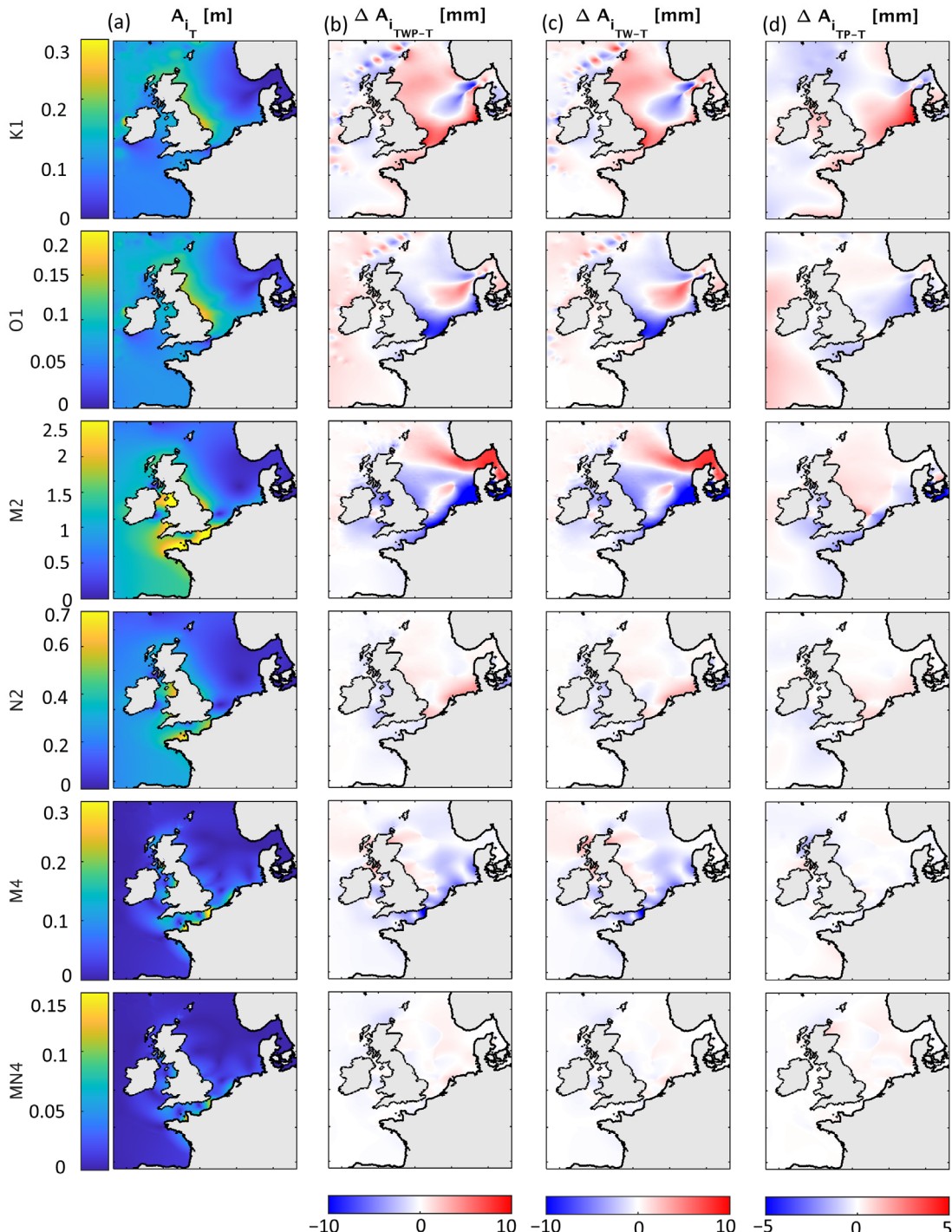

**Figure 4.** Tidal amplitudes and changes for the year 2009, where each row is representative of the studied constituents. Columns show constituent amplitude for the year 2009 (from a tide-only forced model) (**a**), change in constituent amplitude induced by combined wind and pressure effects (**b**), wind-induced amplitude change (**c**), and pressure-induced change (**d**).

Across all studied constituents, the WP influence (Figure 4b) can be seen to have the largest effect in the North Sea (NS) and along the German Bight (GB), showing values of up to +/−10 mm in this area for K1, O1, and M2 and +/−5 mm for N2 and M4, and up to +/−2 mm for MN4. This area of wind and pressure influence on constituent amplitudes can be seen to stretch along the Northwestern French coast, and up to the Dover Strait (DS).

These influences are positive for K1, N2, and MN4, and negative for O1, M2, and M4. Areas of slightly weaker but still notable influence can also be seen in the English Channel (EC).

The results considering only the wind effects are largely the same as the spatial patterns and magnitudes of influence for the combined wind and pressure effect. Offshore, in the Bay of Biscay (BoB), it can be seen that the influence of wind becomes much weaker in this area, most notably for the diurnal constituents.

In contrast to the wind-driven effects, the overall pattern of pressure-driven influence is largely different from that of the combined wind and pressure. Influences are smaller on the shelf, showing changes of up to 5 mm (for K1 in the GB). For all constituents, the pressure-driven effects off the shelf break are larger than those arising from wind influences. One of the most notable examples of this is O1. Off the shelf break in the BoB, wind-driven influences in this area do not exceed 1–2 mm in change, whereas in contrast changes pressure-driven changes can be seen to cause 3–4 mm in change.

Figure 5 (below) shows a boxplot of the distribution of values of influence across the full study domain, computed at a 6 km resolution using the data generated for Figure 4.

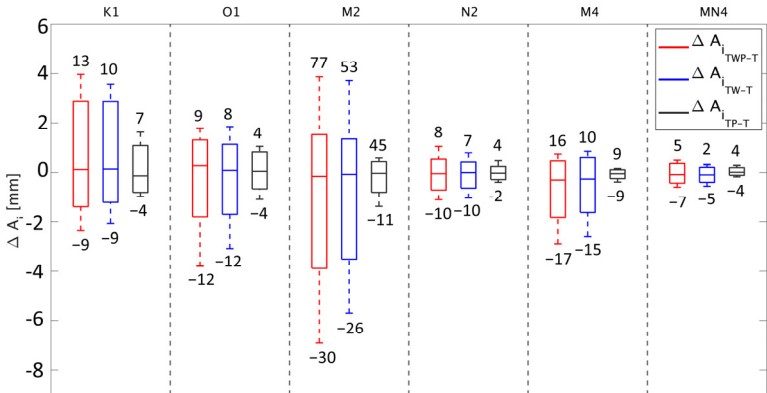

**Figure 5.** Boxplot of tidal constituent amplitude changes induced by the 2009 meteorological forcing, for constituents displayed in Figure 4. Upper and lower bounds of boxes represent the 10th/90th percentiles. Extended lines represent values within 5th/95th percentile. Values above and below extended lines show the maximum and minimum values, respectively.

Firstly, in Figure 5, it is clear that out of the three scenarios, WP exerts the largest range of changes. For diurnal constituents K1 and O1, the 5th/95th percentile values stretch from −2 mm to +4 mm and −4 to +2 mm. Whilst the median WP influence values of these two constituents both sit close to 0 mm, negative effects can be seen to reach lower values in O1 than in K1. This is also true for the influence of W only scenario.

Figure 5 also highlights the influence exerted on M2, where 5th/95th percentile values of −7 mm to +4 mm are seen for WP and −6 to +4 mm for W. If the maximum and minimum effects exerted on M2 are considered, the gap between influences on M2 and the constituent exhibiting the next largest influences (K1) is made even more apparent. The largest positive changes of WP are seen to reach 77 mm on the west coast of Scotland, and influences of up to 53 mm and 45 mm are seen to be exerted by W and P, respectively. The largest negative changes reach as low as −30 mm are found for WP and effects as low as −26 mm and −11 mm are found for W and P, respectively. MN4 is subject to the smallest effects, showing submillimetric 5th/95th WP, W, and P values of influence.

W-driven hydrodynamics induce both a stronger negative and positive change in all constituents than P, except for MN4. P-driven changes in MN4 reach a higher maximum value than W-driven changes (4 mm compared to 2 mm); however, the 95th percentile values for W are still larger than those of P. Considering the close ranges of influences between WP and W, in addition to the matching spatial patterns displayed in Figure 4, it is clear that WP influences are mainly driven by the influence of W.

### 3.2. Sensitivity of 2009 Tidal Amplitudes to Meteorological Conditions (1980–2021)

Figure 6 shows the mean (Figure 6a) and standard deviations (Figure 6b) of effect on tidal constituent amplitudes for TWP scenario considering all years of meteorological forcing between 1980–2021 (using the TWP scenario for the year 2009 as a reference simulation). Figure 6c shows the range of meteorologically induced effects. Figure 6d shows the meteorologically induced evolution of constituent amplitudes at three locations over this period, for three selected points (location indicated in Figure 6a, 1st row) for each year of meteorological forcing. Points P1 and P2 refer to locations selected in the model close to real tide gauge locations (Dunkerque, Santander) while P3 is located in the GB.

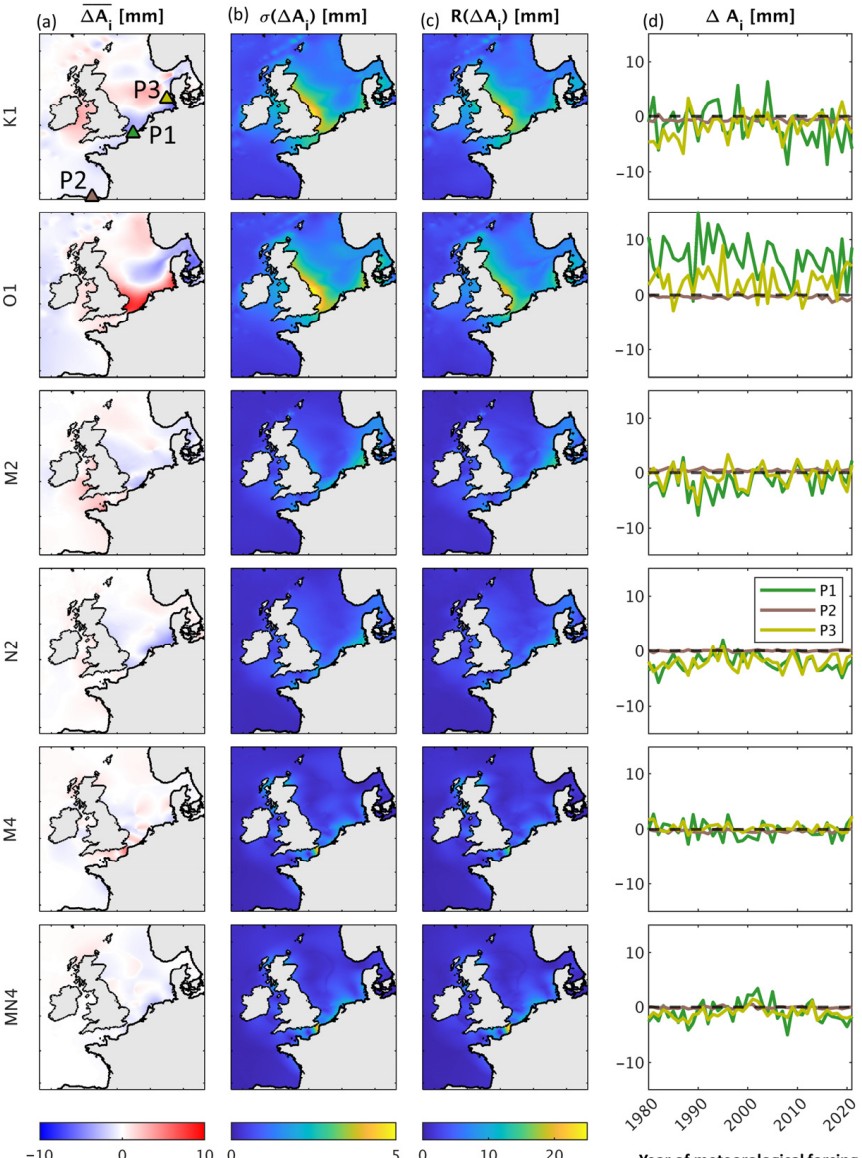

**Figure 6.** Sensitivity of 2009 tidal constituent amplitudes to changing meteorological forcing (years of 1980 to 2021), using 2009 as a reference year. (**a**) Mean change in WP-induced tidal changes over all scenarios (where triangles in K1 show points P1, P2, P3, plotted in Figure 6d). (**b**) Standard deviation of the changes. (**c**) Range of the changes. (**d**) Tidal constituent changes extracted from the three points for each year of meteorological forcing.

K1 (Figure 6a) shows moderate negative mean change in the DS and the GB, showing values of around −2 mm in both areas. An area of positive mean influence can be seen in the Irish Sea, where values of up to +5 mm are reached. In addition, an area of influence

stretching into the NS, from the Eastern English coast is visible. O1 shows stronger mean influences than K1, with values of up to +10 mm seen stretching from the GB, along the coast of Holland, and into the DS. A moderate negative area of influence is seen leaving the Baltic Sea, stretching into the NS, where maximum negative changes of −7 mm are reached. Patterns of mean influence on M2 are relatively weak in comparison to K1 and O1. Both K1 and O1 display the largest areas of variability on the east coast of the UK (Figure 6b,c). In addition to this, comparatively moderate variability can be seen in the GB for both of these constituents. M2 and N2 both show a small negative influence in the GB, where the effect is stronger in N2 than M2 (up to −3 mm in N2, and −1 mm in M2). M2 also shows an area of positive influence in the Western EC that is not present in N2. The largest variability in the semi-diurnal constituents is present in the GB, showing values of standard deviation of up to 2.5 mm. M4 shows a small area of positive influence in the DS, as well as an area of weak negative influence in the GB. Influences on MN4 are small in magnitude and are only visible on the coast of Holland (where a weak negative influence can be seen). Both M4 and MN4 display areas of high variability in the DS. It is obvious that for all constituents, the variability is higher on the shelf.

Figure 4d shows the change for individually selected points for each year of meteorological forcing. The location of P1 for constituent K1 can be seen to oscillate about 0 between the forcing years of 1980–2006, whereafter it can be seen to display a negative mean shift. For O1, the shift is seen to be overall positive, with a mean shift of 7 mm as shown. M2 and N2 are seen to be overall negative, with mean shifts of −2 mm for both. M4 shows an oscillation around 0, similar to the observed for K1 (between 1980 and 2006). Finally, MN4 displays a small negative mean shift of −1 mm considering all years but shows a positive shift between 2000–2003, where a maximum shift of 3 mm has occured in 2002. At the location of P3, similar behaviour to P1 is seen in most instances, except for K1. In the K1 series at the location of P2, there is a constant negative shift across years of forcing, with an overall mean shift of −2 mm. For O1, an overall positive mean shift of 2 mm is seen with negative shifts of −1 mm and −2 mm for M2 and N2, respectively. A submillimetric negative mean shift is seen in M4, and a negative shift of −1 mm is seen for MN4. P2 location shows little overall change, displaying submillimetric mean shifts for all constituents.

### 3.3. Effect on Highest Tide

Figure 7a shows the mean meteorological effect on all the highest annual (predicted) tides considering all meteorological forcing years. Figure 7b shows the range of influences, and Figure 7c shows the maximum absolute effect multiplied by its sign (calculated using Equation (9)).

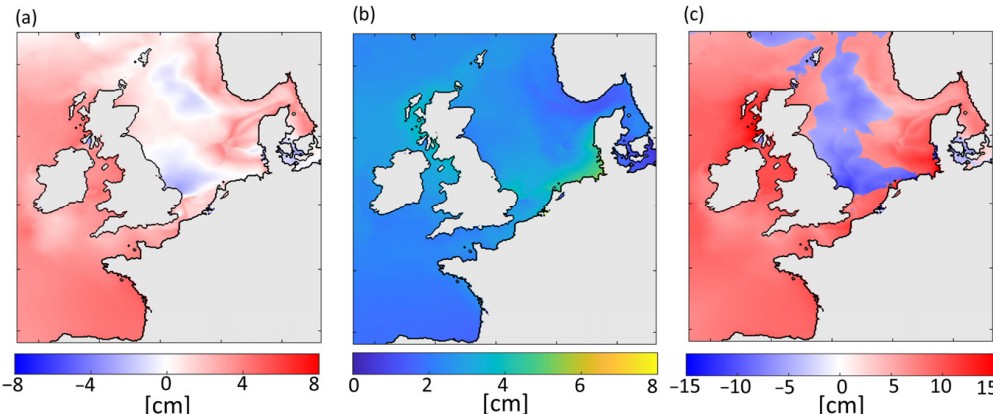

**Figure 7.** Change in highest tide considering all meteorological forcing years for the TWP scenario using the 2009 T scenario as a reference. (**a**) Mean changes in highest tide. (**b**) Standard deviation of changes in highest tide. (**c**) Maximum absolute effect on highest tide taking into account sign of change (see Equation (9)).

Values of mean highest tide change (Figure 7a) can be seen to reach up to 8 cm, with a large positive influence seen off the shelf break in the BoB. Areas of 0 cm mean influence can be seen off the east coast of England and in the northern NS. These areas of no mean influence surround areas of negative mean influence, with values reaching as low as −3 cm.

Standard deviation of highest tide changes (Figure 7b) show the GB to be the highest area of variability, reaching standard deviation values up to 6 cm. Additionally, areas of comparatively moderate variability can be seen on the west coast of Scotland and on the east coast of England (both reaching up to 4 cm). Conversely, areas of lowest variability can be seen in the southwest of Norway, where standard deviation values of up to 1 cm can be seen. Off the shelf break, in the BoB, a widespread area of spatially consistent moderately low variability is present (2 cm).

Maximum absolute changes taking into account sign of change (Figure 7c) are overall largely positive, with the largest positive influence found on the west coast of Scotland (values of around +15 cm). Additionally, an area of positive change similar in magnitude can be seen in GB. The largest area of negative influence can be seen in the NS (values of around −15 cm), where the full extent of the negative influence stretches from the northern coast of Holland up to the northern NS.

### 3.4. Influence of Meteorologically Induced MSL Changes

Through THA, it was found that the meteorological forcing induced a non-uniform change in MSL (Figure 8).

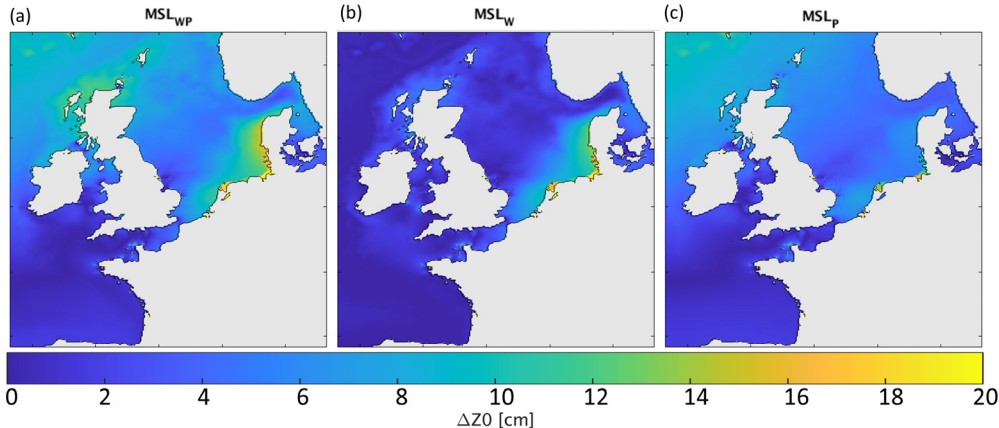

**Figure 8.** Meteorologically induced MSL generated using the meteorological forcing year 2009 for (**a**) the TWP scenario, (**b**) the TW scenario, and (**c**) the TP scenario.

The largest area of MSL-induced by WP ($MSL_{WP}$, Figure 8a) can be seen to occur in the GB, where values reach up to 20 cm. An additional area of notable influence is also discernible off the northwest coast of Scotland, where up to 12 cm of MSL change occurs. Overall, a larger area of influence of WP on MSL is evident on the shelf than off the shelf, with values in the BoB reaching 4 cm.

The analysis of the MSL-induced by W (Figure 8b) and P (Figure 8c), respectively, suggests that the sole pressure induces regional MSL patterns very similar to $MSL_{WP}$, while the wind induces more localised MSL changes, located on the shelf, especially in the GB and along the northwest coast of Scotland. For instance, wind induces MSL changes reaching 8 cm along the northwest coast of Scotland.

Figure 9 below shows this ratio for each of the studied constituents.

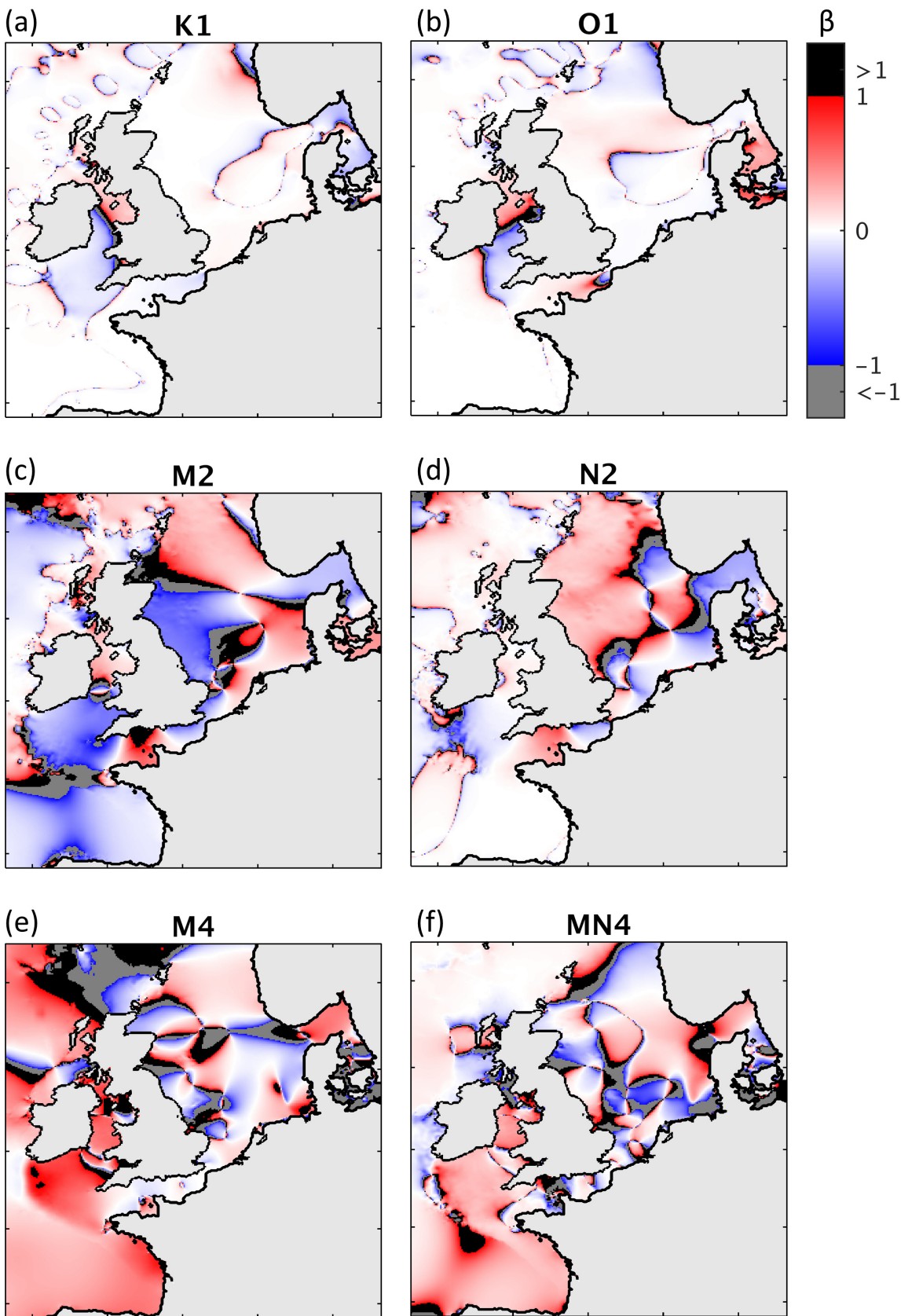

**Figure 9.** Ratio ($\beta$) of the part of MSL-induced change to overall meteorologically induced tidal change for (**a**) K1, (**b**) O1, (**c**) M2, (**d**) N2, (**e**) M4, and (**f**) MN4 (where grey indicates areas where the ratio is larger than one or smaller than zero). See text for details.

Both diurnal constituents (K1, Figure 9a and O1, Figure 9b) show low β across most of the modelled domain, indicating that the influences of wind and pressure-driven changes on diurnal constituents occur largely as a result of factors not taken into consideration by meteorologically induced MSL changes. There are some areas of exception to this, such as in the northern Irish Sea, where β values close to 1 are present (with small areas of β values > 1). Comparably, semi-diurnal constituents (M2, Figure 9c, and N2 Figure 9d) show much larger β ratios. Both show β approaching one in the western section of the English Channel, with negative β values of equal magnitude in the eastern section. Both semidiurnal constituents also show positive β values in the Northeastern North Sea. β ratios for M2 and N2 strongly differ both in terms of patterns and sign. For instance, on the eastern coast of the UK, M2 displays large negative β, whilst N2 displays moderate positive β. The overall prevalence of strong positive β values across both semi-diurnal constituents indicates, in some areas (such as the English Channel), that MSL-driven effects may be the dominant cause of wind and pressure-driven changes, whilst areas such as the Skaggerak strait show that other factors are dominant. Quarter-diurnal constituents (M4, Figure 9e and MN4 Figure 9f) both show β approaching one in the Bay of Biscay. M4 shows predominantly positive β values, with MN4 showing a more varied mix of positive and negative areas.

## 4. Discussion

### 4.1. Causes of Change

The shallow water equations (Equations (13) and (14)) can be expressed as follows (where the horizontal viscosity term $(A\nabla^2\mathbf{u})$ is omitted for clarity):

$$\frac{\partial \zeta}{\partial t} + \nabla\cdot(D\cdot\mathbf{u}) = 0 \tag{13}$$

$$\frac{\partial \mathbf{u}}{\partial t} + \mathbf{u}\cdot\nabla\mathbf{u} - f\mathbf{k}\cdot\mathbf{u} = -g\nabla\zeta + \frac{1}{\rho}\nabla P_a + \frac{\boldsymbol{\tau_s}}{\rho D} - \frac{\boldsymbol{\tau_b}}{\rho D} \tag{14}$$

where $\mathbf{u}$ is depth-integrated current velocity (in these equations, bold scripting is used to indicate vectors), $\zeta$ is the free surface, D is the total water depth (composed of the sum of water depth, H, and $\zeta$), $\rho$ is sea water density, $P_a$ is atmospheric pressure at sea level, f is the Coriolis parameter ($2\omega\sin\phi$, where $\omega$ is the angular speed of Earth's rotation, and $\phi$, the latitude), $\mathbf{k}$ is a unit vector in the vertical direction, and g is gravitational acceleration. $\boldsymbol{\tau_s}$ and $\boldsymbol{\tau_b}$ are the bed and wind shear stress, respectively.

We find that on the shelf (represented in blue in Figure 1), patterns of WP influence are dominated mainly by W (showing the same orders of magnitude and spatial patterns, Figures 4 and 5). In addition, when moving off the shelf, into deeper water (indicated in black, Figure 1), the influence of W drops rapidly with increasing depth, where a larger influence of P can be seen. Indeed, in the momentum equation of the shallow water equations (Equation (14)), the effect of atmospheric pressure does not depend on the water depth, while the wind effect is depth dependent (implying shallower water leads to a larger effect). A deeper analysis of these equations (Equations (13) and (14)) reveals three primary effects arising from non-linear interactions among tides, wind, and pressure [13,19,35]. These effects include advective effects resulting from advective terms in the momentum equation, quadratic parameterization of bottom friction, and the shallow water effect originating from non-linear terms associated with water depth, D (=H + ξ).

As an illustration of the dominant effect of wind on tide changes in some shallow areas, we may notice that many shallower areas that are found to exhibit high wind-driven interactions (specifically in the German Bight and English Channel, Figure 4c) are also found to exhibit larger changes in high tide (Figure 7c).

Comparing our results to Idier et al. [7] who investigated the sea-level rise effect on tides, we find many areas of agreement. For diurnal constituent amplitudes, we find a strong positive β (i.e., a dominant role of the meteorologically induced MSL on tide

changes) in the Northern Irish Sea, which is supported by the results of [7] where this area is found to exhibit a moderate positive relationship between MSL and diurnal constituent amplitude. The positive β values that we find in the western English Channel for semidiurnal constituents are consistent with findings in [7], where it is found that MSL change exerts strong negative influences on semi-diurnal constituent amplitudes in this area. Comparing quarter-diurnal constituents, we see a spatial similarity in the Celtic Sea between positive β values, and moderate positive MSL/constituent amplitude relationships [7].

Research in the field of MSL-induced tidal characteristic changes is plentiful (e.g., [7,9,36]), and any influence attributed to this meteorologically induced MSL may be explained through the same mechanisms through which MSL has been shown to influence tidal characteristics. Idier et al. [7] found that a significant part of the changes could be explained by the reduced damping of tides by bottom friction (e.g., using a reduction in the frictional term corresponding to 2 m sea-level rise, they found high tide changes ranging from −10 to 20 cm). In addition, it was also found that alterations in resonant modes occurring as a result of MSL-induced depth changes were partially responsible for MSL-induced tidal change.

We find a large area of moderate-to-high β in the Bay of Biscay for quarter-diurnal constituents, as in [7], and it is noted that this area exhibits a resonant frequency close to the quarter-diurnal constituent periods. From this, we can infer that resonance may therefore explain the dominance of MSL-induced changes of quarter-diurnal constituents in the Bay of Biscay.

### 4.2. Comparison with Areas of Significant Tide–Surge Interaction

To the best of the authors' knowledge, research into the direct effects of wind and pressure on tides at an annual scale over the European shelf has not been carried out. Therefore, this means we cannot draw direct comparisons between our results and similar studies. However, parallels in spatial influence may be drawn. Areas in which a tide–surge interaction exists imply wind and/or pressure-driven tidal changes in these areas. As is noted in Section 3, one area in which wind (and pressure to a lesser extent) influence can be seen is the Dover Strait (see also Figure 4 to see the dominant effect of wind). Section 1 noted that Dunkerque was highlighted in Idier et al. [13] as an area displaying particularly strong tide–surge interactions. Such strong interactions (and related wind and pressure effects on tides) are directly related to the fact that the Dover Strait is subject to significant wind-induced currents (see, e.g., [37,38], showing non-tidal residual currents up to 0.8 m/s, and able to cancel the tidal current reverse) and that these strong currents are aligned with the main tidal currents direction in a quite shallow area (about 30 m over the major part of the strait). Horsburgh and Wilson [15] also highlighted that GB is subject to high tide–surge interaction, similar to the results presented here.

### 4.3. Comparison with Observed Changes

Results in this paper may also be compared to changes in overall tidal amplitudes. Comparing to previously discussed changes in Section 1, it can be seen that the changes presented here are of a similar magnitude. For example, changes induced by combined wind and pressure effects can be seen to affect semidiurnal constituents by between −3 cm and 7.7 cm (considering the largest WP-induced changes in M2 in 2009, Figure 3). These changes fall within the magnitude of changes discussed in Section 1 (where, for example, M2 amplitudes at Brest were found to decrease by 11 cm between 1890–1945 and increase by 4 cm from 1945–2005 [4]).

### 4.4. Implications for Coastal Risk Assessment

Changes in tidal levels can drastically affect return levels. Pickering et al. [36], for example, calculated that for a 7 cm change in high tidal levels (as a result of a 1 m MSL rise), a 1 in 100 year event, at Xiamen, China, and New Orleans in the United States, became 1 in 63 and 1 in 73 year events, respectively. An analysis of tide gauges along the French coast has shown that changes on a centimetre scale can yield large changes in extreme high water

level return periods. For example, at Brest, a 15 cm change in extreme high water level (from 440 to 455 cm) yields a return period change from 3 to 50 years [39]. In addition, for most coastal applications, extreme value analyses are performed assuming that the population of extremes is stationary. Work has been carried out to develop time-dependent extreme value analysis methods [40,41]. The necessity of these developments comes from identified pluriannual variability in atmospheric surges and waves. This study, however, highlights that such meteorological variability may also induce changes in the tidal part of water level, which is most of the time still considered stationary in this time-dependent extreme value analysis. Therefore, the addition of this part of change to previously developed approaches in time-dependent extreme value calculations may contribute to reducing the uncertainties in the prediction of extremes.

## 5. Conclusions

In this study, the influence of wind and pressure-induced changes on tidal constituent amplitudes and highest tides were investigated through a hydrodynamic modelling approach to establish both their combined and relative contributions. For the year 2009, combined wind and pressure effects were found to affect tidal amplitudes by up to $+/-1$ cm, with larger effects being demonstrated for constituents with larger amplitudes. The wind effect on the amplitude of the tidal constituents is found to be dominant in shallow water, while the pressure effect is dominant in deeper areas (using the example of the shelf break off the Bay of Biscay).

Replacing the hourly meteorological condition of the year 2009 with conditions corresponding to other years, we found that constituent amplitudes change by an additional $+/-1$ cm. Tracking tidal responses of individual locations to changing meteorological forcing, three distinct behaviours were observed to arise and are as follows: areas in which an overall positive mean shift occurs, areas over which an overall negative mean shift occurs, and areas in which little mean shift occurs, but values of change oscillate around 0 mm.

Analysing the largest effect on highest tide, inclusive of sign, the influence of wind and pressure is largely positive with magnitudes that locally reach 15 cm (positive or negative). Maximum changes up to $+/-15$ cm are found along the west coast of Scotland and in the German Bight, where standard deviations on highest tide change also show these areas to exhibit large variability (up to 6 cm) as a result of wind and pressure influence.

The results are interpreted in terms of physical mechanisms, based on the analysis of the shallow water equation terms. First, the results are consistent with the known depth-dependency of the wind term and the depth-independency of the pressure term. Second, we show that in some areas, a significant part of wind and pressure-induced tidal changes (especially the semi-diurnal and quarter-diurnal components) can be explained by the meteorologically induced non-uniform MSL changes. From an analysis of the literature on the effect of MSL changes on tides, we suggest that this meteorologically induced MSL modifies the tides through bottom friction damping and resonant mode shifts.

This work contributes to an ongoing body of research focused on non-astronomical drivers of tidal change and the physical mechanisms driving them. The implications for this study lie within the fields of coastal flooding assessments and extreme water level calculations. This work may be extended to further non-astronomical drivers of tidal change, such as MSL, where it may be estimated for which MSL changes, the effect of the time variability of wind and pressure fields become negligible in comparison.

**Author Contributions:** Conceptualisation, Methodology, and Formal analysis, J.C., D.I., G.W. and G.A.; Writing—original draft, and visualisation, J.C.; writing—review and editing and supervision D.I., G.W. and G.A. All authors have read and agreed to the published version of the manuscript.

**Funding:** This work was carried out as part of the MODMAR PhD project (2020–2023), funded by Bureau de Recherches Géologiques et Minières (BRGM), Service Hydrographique et Océanographique de la Marine (SHOM) and La Rochelle Université.

**Institutional Review Board Statement:** Not applicable.

**Informed Consent Statement:** Not applicable.

**Data Availability Statement:** Data produced in this study are available on request. GESLA3 dataset available at https://gesla787883612.wordpress.com/downloads/ (accessed on 25 August 2023). CFSR dataset available at https://rda.ucar.edu/datasets/ds093.1/ (accessed on 25 August 2023). CFSv2 dataset available at https://rda.ucar.edu/datasets/ds094-1/ (accessed on 25 August 2023).

**Acknowledgments:** The authors would like to thank Rodrigo Pedreros for his contributions in providing the meteorological datasets and additional data processing scripts.

**Conflicts of Interest:** The authors declare no conflict of interest.

## Appendix A

**Table A1.** Study sites used for validation (Section 2.1), where the numbers correspond to those shown in Figure 1.

| Number | Tide Gauge | Number | Tide Gauge | Number | Tide Gauge |
|--------|------------|--------|------------|--------|------------|
| 1 | Workington | 25 | Llandudno | 49 | Roscoff |
| 2 | Dieppe | 26 | Leith | 50 | Santander |
| 3 | Tredge | 27 | Port Bloc | 51 | Port Tudy |
| 4 | Arun platform | 28 | Plymouth | 52 | Brouwershavensche |
| 5 | Den helder | 29 | Texel Nordzee | 53 | Saint Nazaire |
| 6 | Arcachon Eyrac | 30 | Lowestoft | 54 | North Shields |
| 7 | Millport | 31 | Howth | 55 | Cadzland |
| 8 | Huibertgat | 32 | West Kappelle | 56 | Concarneau |
| 9 | Saint Gildas | 33 | Cromer | 57 | Newhaven |
| 10 | Bournemouth | 34 | Barmouth | 58 | Portrush |
| 11 | Terschelling Nordzee | 35 | Sainildas | 59 | Haringvliet |
| 12 | Aranmore | 36 | Bayonne Boucau | 60 | Scheveningen |
| 13 | Wick | 37 | Sandown Pier | 61 | Dover |
| 14 | Dunkerque | 38 | Aberdeen | 62 | Deal Pier |
| 15 | Mumbles | 39 | Imjuiden Buitenhaven | 63 | Roompot Buiten |
| 16 | Tobermory | 40 | Holyhead | 64 | Harwich |
| 17 | Cherbourg | 41 | Castletown Port | 65 | St. Marys |
| 18 | Ilfracombe | 42 | Whitby | 66 | Port Erin |
| 19 | Petten Zuid | 43 | Kinlochbervie | 67 | Licheiland Goeree |
| 20 | Ullapool | 44 | Portpatrick | 68 | Wierumergronden |
| 21 | Euro platform | 45 | Socoa | 69 | Bangor |
| 22 | Les Sables D'olone | 46 | Teignmouth Pier | | |
| 23 | La Rochelle | 47 | Le Havre | | |
| 24 | Le Conquet | 48 | Calais | | |

**Table A2.** All 94 tidal constituents used in TTB, when applying THA.

| | | | | | | |
|---|---|---|---|---|---|---|
| Mm | Mf | Mtm | MSqm | Q1 | O1 | P1 |
| K1 | 2N2 | N2 | M2 | S2 | K2 | M4 |
| Mqm | 3OK1 | MS1 | S1 | SO1 | 2MN2S2 | 2NS2 |
| 3M2S2 | ST1 | OQ2 | MNS2 | MNuS2 | ST2 | ST3 |
| 2MK2 | SNK2 | MSK2 | OP2 | MKS2 | NKM2 | 2SM2 |
| SKM2 | 2SN2 | 2SMu2 | MQ3 | 2MK3 | MO3 | M3 |
| SO3 | MS3 | MK3 | SP3 | S3 | SK3 | K3 |
| 2MNS4 | N4 | 3MS4 | MN4 | MNu4 | SN4 | ML4 |
| NK4 | MS4 | MK4 | 2MSN4 | S4 | SK4 | 3MNK6 |
| 3MNS6 | 4MK6 | 3MNL6 | 4MS6 | 2MN6 | 2MNu6 | 3MSK6 |
| M6 | 3MKS6 | MSN6 | 2ML6 | MSNu6 | MNK6 | 2MS6 |
| 3MLN6 | 2MK6 | MSL6 | 3MSN6 | 3MKN6 | 2SM6 | MSK6 |
| 2M2N8 | 3MN8 | M8 | 2MSN8 | 3ML8 | 3MS8 | 3MK8 |
| 2M2S8 | 2MSK8 | 2M2K8 | | | | |

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
