# Peer review of "Atmospheric Wind and Pressure-Driven Changes in Tidal Characteristics over the Northwestern European Shelf"

_jmse, doi:10.3390/jmse11091701_

Round 1
Reviewer 1 Report
This is a nice paper to discuss the atmospheric wind and pressure-driven changes in tidal characteristics over the Northwestern European shelf. The direction of this paper is novel, but the introduction of this paper and the way of showing the discussions and results can be improved. So, I give the "major revision" to this article. My review comments are below.
1. In the introduction section, the authors should emphasize the objective of this paper clearly and give appropriate descriptions from the literature review.
1.1 Why choose the year 2009 as the yearly case with the meteorological forces? This critical reason should be explained in Introduction.
1.2 I know the tidal characteristics are important to know, but the reference review from Lines 26 to 44 is important and necessary to show?
1.3 The wind and pressure effects on yearly tidal changes are the research focus of the authors' paper. But, why this topic (or this research) is important to explore? I think the authors should give some academic reasons. Saying "no study has focused on the wind and pressure effects" in Lines 60–61 is not enough. For instance, does it affect some phenomenon without discussing this? This information will be more important to the readers.
2. The authors presented Figure 3 as their validation for the amplitudes of each tidal constituent. But MN4 seems to be apart from the diagonal line when the amplitude is small. Any reasons for explaining this? I didn't find any discussion from Lines 151 to 158.
3. The discussions from Lines 162 to 175 follow Figure 2. It will be better to put these before showing Figure 3 for the coherence of this discussion.
4. I suggest the authors re-organize the parts of the results and discussions and combine these parts together. (1) I can only find the results in Sections 3.1, 3.2, and 3.3, but I cannot understand why the authors choose to present these. The reasons for showing the results shall be discussed. (2) If the authors choose to separate Results and Discussions, then no more figures should be added to Discussions. But Figures 8 & 9 are added in Discussions.
5. In Line 425, why pick 1015 hPa as the reference of the atmospheric pressure?
6. Some terminologies and words should be written in lower cases; for example, mean sea level (in Line 47). The authors should check carefully.
N/A
Reviewer 2 Report
Reviewing of the Article ‘Atmospheric wind and pressure-driven changes in tidal characteristics over the northwestern European shelf’ by J. Challis et al. submitted to the Journal of Marine Science and Engineering (Manuscript ID: jmse-2509153).
The authors work on the driving mechanisms of tide constituent changes (including the highest tide) over a long range of time span (e.g., from 1980 to 2021), especially in year 2009. They design numerical experiments by including one and two factors (e.g., tide only, tide + wind, tide + atmospheric pressure, and tide + wind + atmospheric pressure), and they finally found that the wind effects dominant the shelf locally, while the pressure effect mainly in the deep ocean part). The mean sea level (MSL) changes are related to the resonant effect on the quarter-diurnal tide components. I have no doubt on the literature review and writing, since they are all well-constructed and expressed. My suggestion to this Article is Major Revision considering the following major issue.
General comments:
1. The authors have done the model validation with sufficient tide gauges over the studied domain. My major concern is that the root mean square difference (RMSD) of the model-to-data comparison is at several or O (10) cm, while the meteorological effects from other years on year 2009 is only 1 cm. How the model reliability is guaranteed with the factor that the order of model error is higher than the model-induced effect (e.g., ~10 cm versus 1 cm)? Even, the wind and pressure effects are ~15 cm, which is around the model error.
Specific comments:
1. At P1 in L40: ‘ … over the period [of] 1960-2021 … ‘ ;
2. Introduction: Please add more literature review on the study of the effects of wind and pressure on tides.
3. At P2 in L80, 82 and other places: Please add a space between 2km or 6km etc.;
4. At P3 for Figure 1: It may be better adding more marks of Scotland, Norway, France, Dutch, Irish Sea, Celtic Sea, and Dunkerque etc., since these locations were mentioned and discussed in the following contents.
5. At P4 for Figure 2: It is a nice figure.
6. At P7 in L 231 – 235 : ‘In the present MARS model setup … with the S2 constituent obtained from T simulations’. I am not quite understanding why the S2 constituent has to be obtained from the tide-only simulation, but not from the tide-wind-pressure simulations? It is mentioned herein, but I just cannot follow the explanation. Please explain it further to the reviewer. Thanks.
7. At P1 in L 349 : ‘ … between 200-2003, …’. Is that a typo for ‘200’? Please double check it.
8. At P12 Discussion. Please indicate it in the figure or some other ways, by specifying the areas of ‘shelf’.
9. At P12 in L 391: ‘ … in the momentum equation of the shallow water equations (see e.g. [33]) …’. Instead of seeing [33], would it be better adding the equation herein or in the Appendix? So that maybe easier for readers to follow it.
10. At P13 in L 429: ‘ … calculate the (fictional) equivalent pressure field … ‘. Please double check ‘fictional’?
11. At P14 in L 436: what does this sentence mean ‘ … a ratio (beta) between the 2 is calculated’?
12. At P15 in L464: ‘… we find a strong positive beta [in] in the northern Irish Sea …’;
Minor editing of English language required
Reviewer 3 Report
This paper analyzes the contribution of atmospheric factors such as wind and pressure to the tidal amplitude of the continental shelf areas in northwestern Europe by constructing numerical experiments. The calculation results show that the average variations induced by these factors for tidal amplitude is ± 1cm, and the maximum amplitude can reach ± 15cm. The topic of the paper is interesting, the expression is clear, the conclusion is sufficient, and the language is also fluid. I think the paper in its current form has met the publication requirements. I have some minor comments for the author to consider revising the paper.
(1) The amplitudes of tidal changes reach ± 15cm, and this meaning needs to be clarified clearly. Is this an average of 15cm, or is it the 15cm that appears in a certain location or at a certain time? If it is the average of a spatiotemporal range, then the extreme value at a certain position and time may be higher than 15cm. Moreover, it is better to further summarize under what circumstances this 15cm change (i.e., strong responses to atmospheric factors) will occur. For example, is it due to the influence of strong wind or storm surge, and the combination of astronomical tide and storm surge that the water level changes significantly?
(2) The explanation of tidal amplitude changes caused by wind and pressure should be written more clearly in the Conclusion and Abstract sections. In the Conclusion, although it is related to mean sea level, the explanation is not clear and easy to understand. The analysis in Section 4.1 is quite detailed, however, in the Conclusion section this aspect is not well summarized. In the Abstract, there is no mention of the reason for the tidal changes caused by wind and pressure at all. IMO, the understanding of this reason should be a significant point of this study.
Round 2
Reviewer 1 Report
From the revised manuscript, the authors have answered my review comments proposed last time and also explained clearly in their cover letter. So, I give this manuscript "accepted".
Another minor point: Equations 13 and 14 should use /bullet symbols not /period symbols. The authors should be able to correct this and do proofread carefully for the whole manuscript in the final round.
Author Response
The appropriate corrections have been made to equations 13 and 14 (replacing the period symbols with bullet symbols, as per the reviewer's comment). In addition, a thorough proofread has been carried out, paying careful attention to equation formatting.
Reviewer 2 Report
Re-reviewing of the Article ‘Atmospheric wind and pressure-driven changes in tidal characteristics over the northwestern European shelf’ by J. Challis et al. submitted to the Journal of Marine Science and Engineering (Manuscript ID: jmse-2509153).
The authors have addressed my minor comments well and make important efforts to revise the manuscript. However, the model error is still not acceptable for me to trust that the meteorology influences are reasonably estimated (only 1cm compared to ~10 cm model error). Therefore, my suggestion is Major Revision for this article.
